# Factors determining cognitive, motor and language scores in low birth weight infants from North India

**Ravi Prakash Upadhyay**[1,2]*, **Sunita Taneja**[1], **Suman Ranjitkar**[3], **Sarmila Mazumder**[1], **Nita Bhandari**[1], **Tarun Dua**[4], **Laxman Shrestha**[3], **Tor A. Strand**[5]

**1** Centre for Health Research and Development, Society for Applied Studies, New Delhi, India, **2** Department of Global Public Health and Primary Care, University of Bergen, Bergen, Norway, **3** Department of Child Health, Institute of Medicine, Tribhuvan University, Kathmandu, Nepal, **4** Department of Mental Health and Substance Abuse, World Health Organization, Geneva, Switzerland, **5** Department of Research, Innlandet Hospital Trust, Lillehammer, Norway

* ravi.upadhyay@sas.org.in

## Abstract

### Background

Children born with low birth weight (LBW) tend to have lower neurodevelopmental scores compared to term normal birth weight children. It is important to determine factors that influence neurodevelopment in these low birth weight children especially in the first 2–3 years of life that represents a period of substantial brain development.

### Methods

This secondary data analysis was conducted using data from LBW infants enrolled soon after birth in an individually randomized controlled trial (RCT) and followed up till end of 1st year. Neurodevelopmental assessment was done at 12 months of corrected age by trained psychologists using Bayley Scales of Infant and Toddler Development 3rd edition (Bayley-III). Factors influencing cognitive, motor and language scores were determined using multivariable linear regression model.

### Results

Linear growth (i.e., length for age z score, LAZ) [*cognitive*: Standardized β-coefficient = 2.19, 95% CI; 1.29, 3.10; *motor*: 2.41, 95% CI; 1.59, 3.23; *language*: 1.37, 95% CI; 0.70, 2.04], stimulation at home [*cognitive*: 0.21, 95% CI; 0.15, 0.27; *motor*: 0.12, 95% CI; 0.07, 0.17; *language*: 0.21, 95% CI; 0.16, 0.25] and number of diarrhoeal episodes [*cognitive*: -2.87, 95% CI; -4.34, -1.39; *motor*: -2.62, 95% CI; -3.93, -1.29; *language*: -2.25, 95% CI; -3.32, -1.17] influenced the composite scores in all three domains i.e., cognitive, language and motor. While increase in LAZ score and stimulation led to increase in composite scores; an increase in number of diarrhoeal episodes was associated with decrease in scores. Weight for height z scores (WHZ) were associated with motor and language but not with cognitive scores. Additionally, a negative association of birth order with cognitive and language scores was noted.

**Data Availability Statement:** All relevant data are available from figshare (https://figshare.com/s/c932d11ff5101e2268bf).

**Funding:** The original study was funded by Grand Challenges Canada (Grant Number 0725–03) and the Research Council of Norway (RCN) through its Centers of Excellence Scheme (project number 223269). This secondary data analysis was supported by Centre for Intervention Science in Maternal and Child Health (CISMAC), Norway (https://www.uib.no/en/cismac). The funding was recieved by RPU and SR The funders had no role in study design, data collection and analysis, decision to publish, or preparation of the manuscript.

**Competing interests:** The authors have declared that no competing interests exist.

## Conclusions

The findings indicate the possible importance of promoting nutrition and preventing diarrhoea as well as ensuring optimal stimulation and nurturance at home for enhancing child development in LBW infants.

## Introduction

It is estimated that around 250 million (43%) children under the age of 5 years in low- and middle-income countries in 2010 did not reach their full developmental potential [1]. The largest numbers are from Sub-Saharan Africa followed by South Asia [1]. Infants and children with low birth weight (LBW) i.e. with birth weight ≤2500 g have been identified to be particularly at a higher risk of neurodevelopmental impairment. Data from available studies suggest that infants and children born with LBW have a higher risk of lower performance in cognitive, motor and language domains; sub-optimal academic performance and significant behavioural problems and developmental delay compared to term healthy counterparts [2–6].

Globally, 20 million LBW babies are born each year, more than 90% of them represent from developing countries [7]. South Asia alone accounts for around one-fifth of all LBW babies globally [7,8]. It is important to identify factors that influence neurodevelopment in these low birth weight individuals especially in the first 2–3 years of life as this is the period when substantial brain development occurs [9,10]. Such an understanding will not only help putting together interventions that could accelerate development but also intervene early for factors that may negatively influence development. Evidence from published studies done in low-middle-income settings points towards certain biological factors that pose a risk for poor neurodevelopment in children. These risk factors include prematurity, low birth weight, poor nutrition, inadequate linear growth reflected as stunting, early childhood illnesses such as diarrhoea, poor socio-economic status and low maternal education [11–16]. Responsive stimulation during the first years of life has also been shown to be crucial for later cognitive development of children [17]. There are, however, limited studies that study the risk factors in low birth weight infants. It is important to examine whether the risk factors for poor neurodevelopment vary among normal healthy children and those born with low birth weight. With this background, the primary objective of the current analysis was to determine the risk and protective factors for cognitive, language and motor performance in low birth weight infants from rural Haryana, India.

## Materials and methods

### Study design, setting and participants

This secondary data analysis used data from an individually randomized controlled trial (RCT) aimed to evaluate the effect of community-initiated Kangaroo Mother Care (ciKMC) on neurodevelopmental outcomes of infants born low birth weight at 12 months of corrected age [18]. A total of 552 stable late preterm or term infants small for gestational age identified within 72 hours of birth and weighing between 1500–2250 g were included in the trial (ClinicalTrials.gov identifier NCT02631343) [18]. The study was conducted in rural and semi urban population in districts Faridabad and Palwal, Haryana. Details of the study areas and the trial procedures have been published [18].

## Participants and data collection

Pregnant women were identified and followed up by study workers through a 3 monthly household surveillance programme. Babies born at home and also those born in a health facility were included, if KMC was not initiated in the facility. In case of home delivery, visit by the team was made at the earliest in order to weigh the baby. In mothers who delivered at a health facility, study team contacted the mother-infant dyad at home, soon after discharge and screened the child for enrolment and measured weight, if still inside the 72 hours enrolment window. Weight of the infant was measured using digital weighing scale (AWS-SR-20; American Weigh Scales, Cumming, GA, USA).

To be eligible for the study, a newborn had to weigh between 1500–2250 g within 72 hours of delivery. Both singleton and multiple births were eligible. Infants who were unable to feed, those with breathing problems, those with gross congenital malformations, or who were less than normally active were referred to hospitals and therefore, not included in the study [18]. Written informed consent in the local language was obtained from caregivers by a study worker prior to enrolment of participants. The intervention comprised of promotion and support of skin-to-skin contact and exclusive breastfeeding. Mother-infant dyads allocated to the intervention group were visited at home by a trained intervention delivery team to explain and initiate KMC and support its practice. All infants in the intervention and control groups received Home Based Post Natal Care (HBPNC) visits by government health workers (Accredited Social Health Activists; ASHAs) as implemented through the health system [18].

The sample size for the primary trial was calculated based on the expected difference of 0.25 SD (around 3.75 points) for cognitive, language and motor outcomes between the intervention group and the control group infants [18]. Considering a power of 80% and assuming a 10% loss to follow up, a total of 552 infants were enrolled. Ethical clearances were obtained from the Ethics Review Committee, Society for Applied Studies, New Delhi and WHO Ethics Review Committee. State approvals were available through a Memorandum of Understanding signed with the National Health Mission in Haryana.

## Outcomes

The neurodevelopmental assessment was done by psychologists using Bayley Scales of Infant and Toddler Development 3$^{rd}$ edition (Bayley-III). The Bayley-III is a comprehensive assessment tool of developmental functioning in infants and toddlers aged 1–42 months [19]. The test is administered directly with the child, takes 40 to 60 minutes to administer and includes three main subscales: cognitive, language (receptive and expressive) and motor (fine and gross) [19]. The BSID-III was adapted for use in the study setting. Details of the adaptation have been provided elsewhere [18]. The psychologists were trained and standardized in administering relevant questionnaires and scales. Periodic re-training and standardization exercises were conducted.

## Exposures and other variables

At enrolment, information was collected on household characteristics (social class, religion, wealth quintile, type of family), infant and birth characteristics (sex, place of delivery, type of delivery, gestational age, birth order, parity and timing of initiation of breastfeeding) and maternal and paternal characteristics (age, education and paternal occupation). Gestational age was assessed either through documented records/available ultrasound reports or through self-reported date of last menstrual period (LMP). Gathering information on vital status, breastfeeding and morbidity (including any hospitalization) along with anthropometric measurements (weight and length) was done by an independent trained outcome ascertainment team during their home visits at child age 1, 3, 6 and 12 months. Caregivers were asked about morbidities in

the previous 2 weeks from the visit. Diarrhoea was operationally defined as three or more loose stools in a day and pneumonia was defined as presence of difficult and fast breathing.

Data on maternal depressive symptoms was assessed using the Patient Health Questionnaire (PHQ)-9 [20]. The 9 items of PHQ-9 tool are based on the DSM-IV diagnostic criteria and higher scores reflect presence of depressive symptoms. To measure infant temperament, infant temperament scale was used as adapted in the MAL-ED study [21]. This 47 item-scale covered six domains i.e. activity, positive emotionality, negative emotionality, sociability, attention and soothability where higher scores reflect more difficult temperament. Maternal sense of competence was assessed using "maternal self-efficacy scale" that consists of 10 questions with four-point scale responses: higher scores reflecting better maternal self-efficacy [22]. Maternal postnatal attachment scale was used to assess mother-infant bonding. This scale consists of 19 items with higher scores reflecting better bonding [23]. PROCESS (Pediatric Review of Children's Environmental Support and Stimulation) questionnaire was used to assess the intensity of stimulation at home. It consisted of three components: clinical observation, parent questionnaire and toy list. Higher scores reflect better stimulation and support to infant [24].

## Statistical analysis

All analysis was done using STATA version 16.0. The normality of the data was examined using histogram, skewness and kurtosis coefficient. Mean (SD; standard deviation) or median (IQR; inter-quartile range) were calculated for continuous variables depending upon the nature of distribution and proportions for categorical variables. Distribution of baseline characteristics was presented. The composite BSID-III scores for cognitive, motor and language domains were calculated.

Potential variables for inclusion in the univariate model were determined from previous literature. Out of these variables, some were selected for inclusion in the final prediction model based on the method suggested by Hosmer and Lemeshow (Applied logistic regression, Second Edition) [25]. For each of the three outcomes i.e., cognitive, language and motor composite scores, univariable linear regression analysis was done as the initial step. Potential variables were put in the univariate model and their corresponding p-value was recorded. Two sets of variables were created- one set containing variable with p-value <0.20 and the other with variable having a p-value ≥0.20 on univariate linear regression. All those variables for which the P-value was less than 0.20 were considered for the multivariable regression analysis. Using stepwise backward method, multivariable regression was done wherein all variables for which p-value was greater than 0.05 in the model were removed and the regression model was re-run. This process of model building was continued till the final sets of variables in the multivariate model were statistically significant. In the next step, variables that were excluded based on their p-value of ≥0.20 on univariable regression or a p-value of ≥0.05 in multivariable model were again put in the multivariable regression model one by one. Those that became statistically significant i.e. p<0.05 were retained in the model. The final multivariate model consisted of variables that were statistically significant with p-value of <0.05. Generalized additive model (GAM) plots were generated using R version 3.1.2 (The R Foundation for Statistical Computing, Vienna, Austria) [26]. Plots were created using *mgcv package*. The aim of the GAM plots was to explore nonlinear associations between continuous variables significant in the multivariate model and the composite cognitive, language and motor scores.

## Results

Baseline characteristics of the infants enrolled in the primary trial (N = 552) as well as those available for neurodevelopmental assessments at 12 months of corrected age (N = 516) have

been presented in Table 1. Majority of the infants included in this analysis were born to Hindu families (81.9%) and resided in joint families (73.8%). Mean (SD) birth weight (in grams) and gestational age (in weeks) was 2058.7 (165.3) grams and 35.7 (1.9) respectively. Out of the total infants enrolled, around three-fifth (59.7%) were females. Majority of the infants were of birth order three or less (82.0%).

## Determinants of cognitive scores

Mean (SD) composite cognitive score in the sample was 102.1 (11.8). Findings of the univariable regression has been presented in S1 Table. Findings of the multivariable linear regression have been presented in Table 2. For each unit increase in length for age z score (LAZ), there was around 2 unit increase in composite cognitive score (adjusted $b$ = 2.19, 95% CI; 1.29, 3.10). PROCESS score was significantly associated with cognitive score at 12 months of age (adjusted $b$ = 0.21, 95% CI; 0.15, 0.27). With each episode of diarrhoea, there was around 3-point lower composite cognitive score (adjusted $b$ = -2.87, 95% CI; -4.34, -1.39). Infants with birth order of $\geq$4 had around 4 points lower cognitive score compared to those with birth order of 1 (adjusted $b$ = -3.79, 95% CI; -6.53, -1.05). When birth order was included as a continuous variable in the model, there was a significant association with cognitive scores i.e., with each unit increase in birth order, the cognitive composite scores reduced by 0.95 unit (adjusted $b$ = -0.95, 95% CI; -1.57, -0.32). These four significant variables in the multivariable linear regression model explained around 22.8% of the variation in composite cognitive score. Fig 1 shows the GAM plots. The plots re-affirm the associations observed on multivariable linear regression.

## Determinants of language scores

Mean (SD) composite language score in the sample was 84.9 (9.1). Findings of the univariable regression has been presented in S1 Table. In the multivariable regression model, each unit increase in LAZ score (adjusted $b$ = 1.37, 95% CI; 0.70, 2.04) and weight for height Z score (WHZ) (adjusted $b$ = 0.91, 95% CI; 0.16, 1.65) were associated with increase composite language score (Table 2). With each unit increase of PROCESS score, the composite language score increases by 0.21 units (95% CI, 0.16, 0.25). Compared to infants with birth order of 1, those with birth order of 2 to 3 (adjusted $b$ = -1.69, 95% CI; -3.17, -0.19) and $\geq$4 (adjusted $b$ = -2.62, 95% CI; -4.63, -0.61) had lower language score (Table 2). When birth order was included as a continuous variable in the model, there was a significant association with language scores i.e., with each unit increase in birth order, the composite scores reduced by 0.61 unit (adjusted $b$ = -0.61, 95% CI; -1.06, -0.15). Each episode of diarrhoea led to a 2.25 unit decrease in composite language scores (95% CI, -3.32, -1.17). These variables explained around one-third i.e. 31.1% of variability in the language scores. Fig 2 shows the GAM plots which re-affirm the associations observed on multivariable linear regression.

## Determinants of motor scores

Mean (SD) composite motor score in the sample was 90.2 (10.4). Findings of the univariable regression has been presented in S1 Table. In the multivariable regression model, each unit increase in LAZ score, WHZ score and PROCESS score was associated with an increase in the composite motor score by 2.41 points (95% CI; 1.59, 3.23), 1.09 points (95% CI; 0.19, 2.00) and 0.12 (95% CI; 0.07, 0.17) respectively (Table 2). Each episode of diarrhoea was associated with a decrease in the motor score by 2.62 points (95% CI; -3.93, -1.29). Overall, these variables explained around 19% of the variability in the composite motor scores. Fig 3 shows the GAM plots which re-affirm the associations observed on multivariable linear regression.

**Table 1. Baseline characteristics of the primary trial population and the sample included in this secondary data analysis.**

| Variables | Number (%) (N = 552) | Number (%) (N = 516) |
|---|---|---|
| **HOUSEHOLD CHARACTERISTICS** | | |
| **Quintiles** | | |
| 1 (Least poor) | 110 (19.9) | 109 (21.1) |
| 2 | 112 (20.4) | 105 (20.3) |
| 3 | 110 (19.9) | 99 (19.2) |
| 4 | 110 (19.9) | 100 (19.4) |
| 5 (Poorest) | 110 (19.9) | 103 (20.0) |
| **Religion** | | |
| Hindu | 450 (81.5) | 423 (81.9) |
| Muslim | 98 (17.8) | 89 (17.3) |
| Others[¶] | 4 (0.7) | 4 (0.8) |
| **Social class[£]** | | |
| General | 137 (24.8) | 133 (25.8) |
| Other Backward Class (OBC) | 181 (32.8) | 167 (32.4) |
| Scheduled Caste/Tribe (SC/ST) | 234 (42.4) | 216 (41.8) |
| **Type of family** | | |
| Nuclear | 140 (25.4) | 135 (26.2) |
| Joint | 412 (74.6) | 381 (73.8) |
| **MATERNAL AND PATERNAL CHARACTERISTICS** | | |
| Mean maternal age (years; SD) | 23.0 (3.7) | 23.1 (3.8) |
| Median years of education of mother (IQR) | 5 (0–9) | 5 (0–9) |
| **Mother's occupation** | | |
| Employed outside home | 10 (1.8) | 9 (1.7) |
| Home maker | 542 (98.2) | 507 (98.3) |
| Mean father's age (years; SD) | 26.4 (4.8) | 26.4 (4.7) |
| Median years of education of father (IQR) | 8 (5–12) | 8 (5–12) |
| **Father's occupation** | | |
| Employed in a government/private firm | 225 (40.8) | 212 (41.1) |
| Daily wage earner | 126 (22.8) | 116 (22.5) |
| Self-employed (own business/farming) | 177 (32.1) | 166 (32.1) |
| Unemployed | 24 (4.3) | 22 (4.3) |
| **BIRTH RELATED CHARACTERISTICS** | | |
| **Place of delivery** | | |
| Home | 160 (29.0) | 148 (28.7) |
| Government facility | 283 (51.3) | 266 (51.5) |
| Private facility | 109 (19.7) | 102 (19.8) |
| **Type of delivery** | | |
| Normal | 546 (98.9) | 511 (99.0) |
| Caesarean section | 6 (1.1) | 5 (1.0) |
| **Birth order** | | |
| 1 | 206 (37.3) | 191 (37.0) |
| 2–3 | 244 (44.2) | 232 (45.0) |
| ≥4 | 102 (18.5) | 93 (18.0) |
| **Parity** | | |
| Primiparous | 206 (37.3) | 191 (37.0) |
| Multiparous | 346 (62.7) | 325 (63.0) |
| **INFANT CHARACTERISTICS** | | |

*(Continued)*

**Table 1.** (Continued)

| Variables | Number (%) (N = 552) | Number (%) (N = 516) |
|---|---|---|
| **Sex of the baby** | | |
| Male | 229 (41.5) | 208 (40.3) |
| Female | 323 (58.5) | 308 (59.7) |
| Mean birth weight (grams; SD) | 2058.6 (167.0) | 2058.7 (165.3) |
| Mean gestational age (weeks; SD) | 35.6 (1.9) | 35.7 (1.9) |
| Early initiation of breastfeeding (within an hour of birth) present | 341 (61.8) | 323 (62.6) |
| Exclusive breastfeeding at 3 months* | 250 (48.7) | 239 (48.4) |

ǥOthers: Christian/Sikh/Jain/Parsi/Zoroastrian/Buddhist/neo Buddhist

£General- group that do not qualify for any of the positive discrimination schemes by Government of India (GOI), OBC- term used by the Government of India to classify castes which are socially and educationally disadvantaged, SC/ST- official designations given to groups of historically disadvantaged indigenous people in India

*data available for 513 infants among the overall study infants and for 494 infants in the sample for this analysis; SD- standard deviation; IQR- Inter-quartile range.

## Discussion

The current analysis was done to examine the factors that may determine cognitive, motor and language scores in low birth weight infants at 12 months of corrected age. We found that linear growth, child stimulation at home and number of diarrhoeal episodes were associated with all three domains of Bayley-III i.e. cognitive, language and motor. While increase in linear growth (represented by LAZ score) and child stimulation (indicated by PROCESS scores) was associated with increase in composite scores; an increase in number of diarrhoeal episodes was associated with decrease in scores. Additionally, we observed a negative association of birth order with cognitive and language scores.

Studies from Asia and Africa have shown association between anthropometric measures of growth with neurodevelopmental outcomes in children, especially the length for age–Z scores [27–29]. In a recent study conducted among young north Indian children, linear growth, child

**Table 2. Determinants of cognitive, language and motor scores, using multivariable linear regression, at 12 months of age in low birth weight infants from rural Haryana (N = 516).**

| Variables | Cognitive score† | | Language score † | | Motor score ǥ | |
|---|---|---|---|---|---|---|
| | *Standardized ß-coefficient (95% CI)* | P-value | *Standardized ß-coefficient (95% CI)* | P-value | *Standardized ß-coefficient (95% CI)* | P-value |
| WHZ at 12 months | ----- | ----- | 0.91 (0.16, 1.65) | 0.017* | 1.09 (0.19, 2.00) | 0.018* |
| **Birth order** | | | | | | |
| 1 | Ref | | Ref | | | |
| 2–3 | -1.51 (-3.56, 0.54) | 0.024*£ | -1.69 (-3.17, -0.19) | 0.018*£ | ----- | ----- |
| ≥4 | -3.79 (-6.53, -1.05) | | -2.62 (-4.63, -0.61) | | | |
| LAZ at 12 months | 2.19 (1.29, 3.10) | <0.001* | 1.37 (0.70, 2.04) | <0.001* | 2.41 (1.59, 3.23) | <0.001* |
| PROCESS Score at 12 months | 0.21 (0.15, 0.27) | <0.001* | 0.21 (0.16, 0.25) | <0.001* | 0.12 (0.07, 0.17) | <0.001* |
| No. of episodes of "Diarrhoea" | -2.87 (-4.34, -1.39) | <0.001* | -2.25 (-3.32, -1.17) | <0.001* | -2.62 (-3.93, -1.29) | <0.001* |

†Adjusted $R^2$ of the multivariable model = 22.8%

† Adjusted $R^2$ of the multivariable model = 31.1%

ǥ Adjusted $R^2$ of the multivariable model = 18.9%

* denotes statistical significance

£P-value calculated based on likelihood ratio.

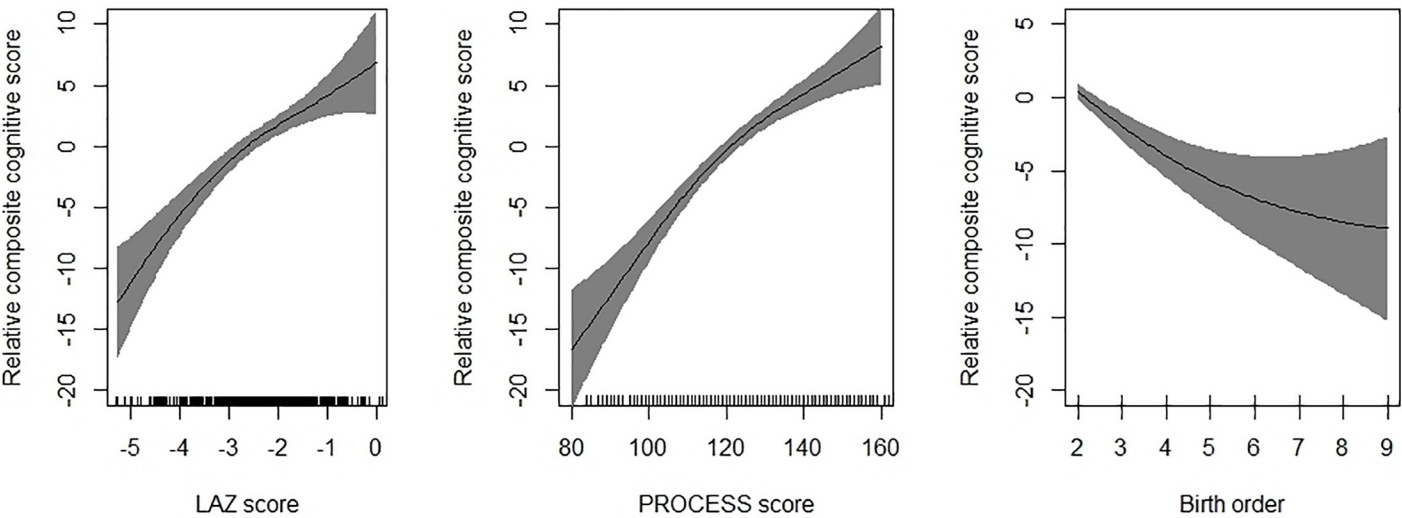

**Fig 1. Association between length for age z-scores (LAZ); PROCESS score and birth order with change in composite cognitive score.** The graphs were constructed using generalized additive models in R, the solid line depicts the association of composite cognitive score and HAZ/PROCESS score/birth order. The Y-axis is centered on the mean composite cognitive score. The shaded area spans the 95% confidence interval of this association.

stimulation and diarrhoeal episodes were associated with scores obtained on Ages and Stages Questionnaire- 3[rd] edition (ASQ-3) [30]. Owing to the overlap in the factors that are responsible for poor growth and neurodevelopment in early childhood, such as inadequate nutrition; morbidities and insufficient care and stimulation, linear growth and neurodevelopment seem to be often closely associated [31]. These adversities lead to deficits in neuronal growth and connectivity within regions of the brain associated with memory, high order learning and motor functioning [32,33]. Stunting, reflecting malnutrition, is a commonly used indicator to identify children with higher risk of developmental deficits [1,34]. We have shown in our analysis that LAZ is a significant determinant of neurodevelopment and support its use as a proxy for neurodevelopment during infancy.

We observed that weight for height z scores (WHZ) were associated with motor and language outcomes. It could be because children who do not experience acute malnutrition possibly have an enhanced capability to engage with the surrounding environment which is

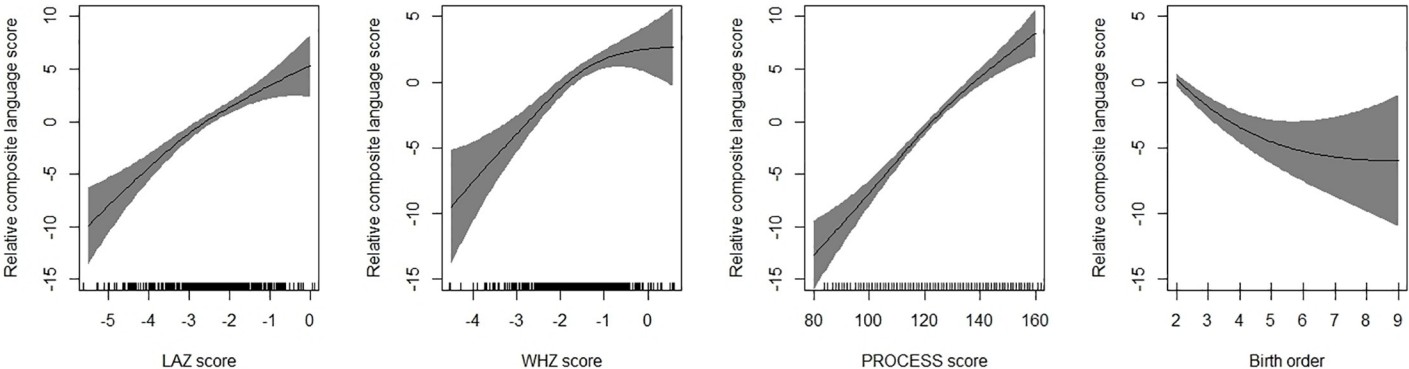

**Fig 2. Associations between height for age z-scores (HAZ); weight for height z-scores (WHZ), birth order, PROCESS score and change in composite language score.** The graphs were constructed using generalized additive models in R, the solid line depicts the association of composite language score and HAZ/WHZ/birth order/PROCESS score. The Y-axis is centered on the mean composite language score. The shaded area spans the 95% confidence interval of this association.

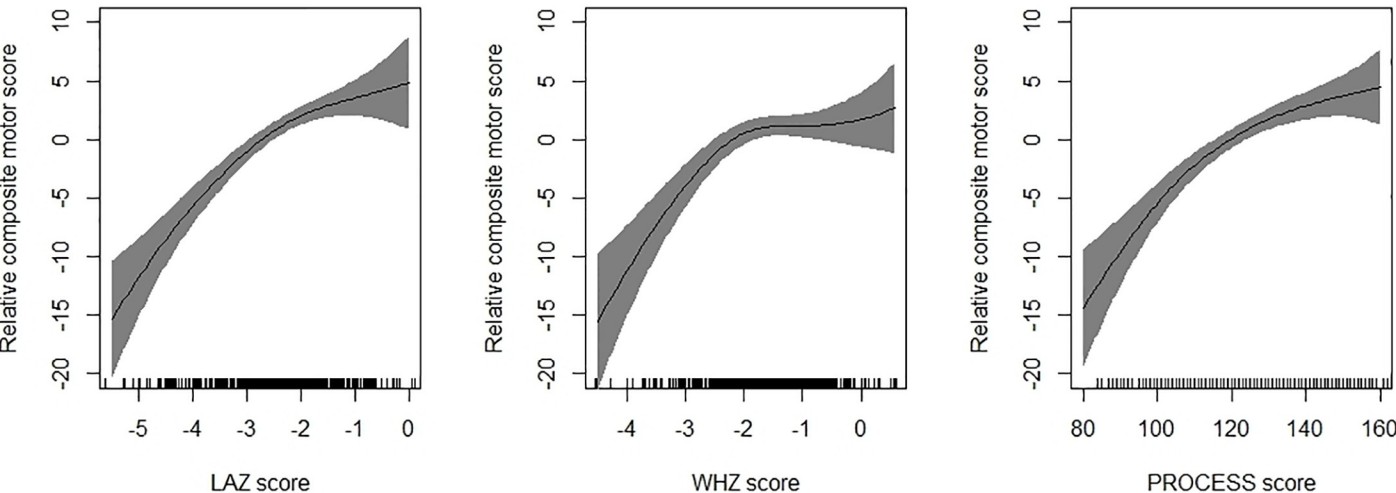

**Fig 3. Associations between height for age z-scores (HAZ); weight for height z-scores (WHZ), PROCESS score and change in composite motor score.** The graphs were constructed using generalized additive models in R, the solid line depicts the association of composite motor score and HAZ/WHZ/PROCESS score. The Y-axis is centered on the mean composite motor score. The shaded area spans the 95% confidence interval of this association.

essential for development [35,36]. Studies have found an association of WAZ with cognitive outcomes; however, we did not find any significant association [35,37]. Our findings are similar to a prospective study in Peruvian children that noted no significant association between weight for length z scores (WLZ) during infancy and intelligence at child age 9 years. [38]. It may be worth to explore whether there is, in real, any differential effect of acute malnutrition on brain functioning.

Similar to our findings, previous studies have also documented a significant negative effect of diarrhoea on neurodevelopmental outcomes [30,38–41]. There are indications that this could be both through its negative impact on growth and its independent effect on gut environment. Intestinal damage as a result of diarrhoea could hamper absorption of critical growth promoting nutrients, thereby affecting linear growth and in turn cognition [41]. On the other hand, occurrence of diarrhoea may instigate systemic inflammatory signals and release of potential neural damaging cytokines [42,43]. Another concept is that of "functional isolation" wherein a child with diarrhoea may be irritable and therefore receive less quality responsive care from the caregivers [44]. In concordance with previous studies showing that children with higher birth order positions are at a disadvantage in terms of cognitive development and educational attainment, we found a negative association of birth order with cognitive and language scores [45–47]. A plausible explanation to justify this association comes from the quantity and quality dilution hypothesis [47]. According to the dilution model, children who are early in the birth order often get a reasonably larger proportion of the family resources or the quality of care received is often better than those who are later in the birth order (who probably have to face more competition). This is considered beneficial for the development of children born early in the birth order [47].

The findings of our study provide impetus to focus on reduction of stunting during infancy, possibly through improved nutrition. The findings also lay emphasis on promotion of early child stimulation interventions and diarrhoea prevention and management during infancy. The study identifies important factors that influence cognitive, language and motor outcomes in low birth weight infants during infancy and could serve as indicators to identify those that need additional care and support.

## Strengths and limitations

The modest sample size of around 500 infants is one of the main strengths. Trained and standardized psychologists conducted the Bayley-III assessments and collected data on stimulation practices at home. The standardized and reliable measurement of anthropometry by trained study team members is also a strength. Another strength is the robustness of the multivariable model for the outcomes considered in the analysis, as reflected by the $R^2$ values. One key limitation is that the study subjects for this analysis were largely stable late preterm or term infants small for gestational age and were only slightly low birth weight. Therefore, the findings may only be applicable to this specific subset of infants. However, for this subset of LBWs, the findings could be generalizable to other South-east Asian and low-middle-income countries as the burden of low birth weight, risk factors for poor development and the quality of care provide to these vulnerable infants is similar in these settings. It should also be noted from Fig 3 that the association between WHZ and composite motor scores may not be linear. Therefore, it would have been more appropriate to include WHZ as quadratic or logarithmic or exponential term or in other transformed form to achieve linearity for regression analysis. However, because such terms are difficult to interpret and communicate, we chose to include it as linear term in addition to showing the shape of association in Fig 3.

## Conclusions

The findings support promoting nutrition and ensuring optimal stimulation at home for advancing child development in infants born with low birth weight. Our study also lends support for diarrhoea disease control as it is an important risk factor for poor neurodevelopment in these vulnerable infants.

## Supporting information

**S1 Table. Findings on univariable linear regression for cognitive, language and motor scores at 12 months of infant age in low birth weight infants from rural Haryana.**
(DOC)

## Author Contributions

**Conceptualization:** Ravi Prakash Upadhyay, Sunita Taneja, Suman Ranjitkar, Tor A. Strand.

**Data curation:** Ravi Prakash Upadhyay, Sunita Taneja.

**Formal analysis:** Ravi Prakash Upadhyay, Sunita Taneja, Suman Ranjitkar, Tor A. Strand.

**Funding acquisition:** Ravi Prakash Upadhyay, Suman Ranjitkar.

**Investigation:** Ravi Prakash Upadhyay, Tor A. Strand.

**Methodology:** Ravi Prakash Upadhyay, Tor A. Strand.

**Project administration:** Sunita Taneja, Sarmila Mazumder, Nita Bhandari.

**Resources:** Sunita Taneja, Sarmila Mazumder, Nita Bhandari.

**Supervision:** Sarmila Mazumder, Nita Bhandari, Tarun Dua.

**Writing – original draft:** Ravi Prakash Upadhyay, Tor A. Strand.

**Writing – review & editing:** Ravi Prakash Upadhyay, Sunita Taneja, Suman Ranjitkar, Sarmila Mazumder, Nita Bhandari, Tarun Dua, Laxman Shrestha, Tor A. Strand.

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
