## [Decision Letter · Decision Letter 0]

26 Feb 2021

PONE-D-20-33924

Factors determining cognitive, motor and language scores in low birth weight infants from North India

PLOS ONE

Dear Dr. Upadhyay,

Thank you for submitting your manuscript to PLOS ONE. After careful consideration, we feel that it has merit but does not fully meet PLOS ONE’s publication criteria as it currently stands. Therefore, we invite you to submit a revised version of the manuscript that addresses the points raised during the review process.

We look forward to receiving your revised manuscript.

Kind regards,

Kannan Navaneetham, PhD

Academic Editor

PLOS ONE

Journal Requirements:

- https://doi.org/10.1016/j.jpeds.2020.05.043

- https://doi.org/10.3945/jn.115.215996

In your revision ensure you cite all your sources (including your own works), and quote or rephrase any duplicated text outside the methods section. Further consideration is dependent on these concerns being addressed.

Reviewers' comments:

Reviewer's Responses to Questions

**Comments to the Author**

1. Is the manuscript technically sound, and do the data support the conclusions?

Reviewer #1: Yes

Reviewer #2: Yes

2. Has the statistical analysis been performed appropriately and rigorously? 

Reviewer #1: Yes

Reviewer #2: Yes

3. Have the authors made all data underlying the findings in their manuscript fully available?

Reviewer #1: Yes

Reviewer #2: No

4. Is the manuscript presented in an intelligible fashion and written in standard English?

Reviewer #1: Yes

Reviewer #2: Yes

5. Review Comments to the Author

Reviewer #1: The study presents the results obtained with a poorly researched population, which is a strength. The main aim of the study is to investigate the factors influencing neurodevelopmental outcomes at 12 months of age in a sample of 516 children from North India. To measure neurodevelopment, the authors use the Bayley-III scales, and they investigate the effect of a significant number of possible factors using a multivariate linear regression model. The results obtained are clearly presented and reveal the positive effect on cognitive, language and motor development of the z-score of length for age (linear growth) and stimulation at home, and the negative effect of the number of episodes of diarrhea. The weight for height z-scores were also positively associated with the composite language and motor scores, but not with the composite cognitive score.

The discussion of the results is adequate and well interpreted in relation to other investigations, and the conclusions that are established are adequate, and well founded on the data.

One aspect that the authors should point out more clearly is that the sample they investigate is only of slightly low birth weight. It would be interesting to extend the study with a sample of very or extremely low birth weight children. On the other hand, it would have been desirable to compare the results of the sample studied to a control group of children with normal birth weight.

Reviewer #2: Abstract:

- please include some numerical results in the 'results' section, e.g. where there was evidence of association between an exposure and outcome, the magnitude of the estimated difference or ratio with its 95% confidence intervals and p-value.

Introduction

- the last sentence on line 78 best belongs in the methods; indeed it is already stated in the first paragraph from line 83, so should probably be removed from here.

Methods

- line 91: a better title would be 'participants and data collection'. It would be helpful to add to this section how the size of the primary trial was determined.

- line 115: a better title would be 'outcomes'

- line 126: a better title would be 'exposures and other variables'. These suggested titles are similar to the sub-sections of the reporting guidelines. Please ensure, having used these titles, that the information reported under each section matches the titles, and move around the ones that don't.

Statistical analysis

- it would be better if the authors presented a brief summary of the potential conceptual relationships between the variables explored for adjustment and (1) the outcome (2) the main exposures, rather than wholly relying on a purely data-driven variable selection approach such as described in line 160. In this summary, the authors should consider the potential directions of the associations between the variables and each of the outcome and main exposure. The reason being that adjustment should only be undertaken for variables that meet the authors' criteria for independent association with the outcome and with the main exposure (e.g. those described in lines 165-167) and not on the causal pathway between the exposure and outcome.

- it is generally more appropriate to report likelihood ratio or other global p-values for categorical variables with more than two levels, such as birth order, than the Wald test p-value for each level. I'm also wondering whether it might have been better to fit birth order as an ordinal explanatory variable, i.e. estimate the effect per unit increase rather than categorise it; did the authors explore this (if so, how)?

- given the shapes of the associations between relative cognitive scores and relative composite language scores with LAZ scores, WHZ scores, PROCESS scores, birth order, it may be worth exploring any departures from linearity in these associations, e.g. by testing whether quadratic terms for these exposures (in addition to the linear term) are significant in the 'final' models, and if so, perhaps incorporate these terms early on in the adjusted model building process.

6. PLOS authors have the option to publish the peer review history of their article (what does this mean?). If published, this will include your full peer review and any attached files.

Reviewer #1: No

Reviewer #2: No

---

## [Author Response · Author response to Decision Letter 0]

21 Apr 2021

We thank the reviewers and the Editor for the valuable comments. We believe that the manuscript has immensely improved

---

## [Editor Report · Decision Letter 1]

26 Apr 2021

Factors determining cognitive, motor and language scores in low birth weight infants from North India

PONE-D-20-33924R1

Dear Dr. Upadhyay,

We’re pleased to inform you that your manuscript has been judged scientifically suitable for publication and will be formally accepted for publication once it meets all outstanding technical requirements.

Kind regards,

Kannan Navaneetham, PhD

Academic Editor

PLOS ONE
---

## [Editor Report · Acceptance letter]

3 May 2021

PONE-D-20-33924R1 

Factors determining cognitive, motor and language scores in low birth weight infants from North India 

Dear Dr. Upadhyay:

I'm pleased to inform you that your manuscript has been deemed suitable for publication in PLOS ONE. Congratulations! Your manuscript is now with our production department. 

Kind regards, 

on behalf of

Prof. Kannan Navaneetham 

Academic Editor

PLOS ONE